# Comparison of Virulence between Two Main Clones (ST11 and ST307) of *Klebsiella pneumoniae* Isolates from South Korea

**DOI:** 10.3390/microorganisms10091827

**Published:** 2022-09-13

**Authors:** Yun Young Cho, Jee Hong Kim, Hyunkeun Kim, Junghwa Lee, Se Jin Im, Kwan Soo Ko

**Affiliations:** 1Department of Microbiology, Sungkyunkwan University School of Medicine, Suwon 16419, Korea; 2Department of Precision Medicine, Graduate School of Basic Medical Science, Sungkyunkwan University School of Medicine, Suwon 16419, Korea; 3Department of Immunology, Sungkyunkwan University School of Medicine, Suwon 16419, Korea

**Keywords:** *Klebsiella pneumoniae*, virulence, serotype, serum resistance, macrophage internalization

## Abstract

In this study, we investigate the characteristics of two main clones of carbapenemase-producing *Klebsiella pneumoniae* isolates from South Korea, ST11 and ST307, including carbapenem-susceptible isolates. Antibiotic susceptibility, serotype or *wzi* allelic type, the presence of virulence genes, and virulence with respect to serum resistance and macrophage internalization were determined for ST11 and ST307 isolates. ST11 isolates had a wide range of characteristics, including serotype and virulence, compared with those of homogeneous ST307 isolates. The *wzi14* or K14 type had higher virulence than that of other serotypes among the ST11 isolates, and the homogeneous ST307 isolates showed similar virulence level as that of the *wzi14*-type ST11 isolates. Our data suggest that it is necessary to monitor not only the introduction and spread of a specific clone, but also its detailed serotype.

## 1. Introduction

Carbapenem resistance by carbapenemase has caused antibiotic treatment failure and has threatened global public health in *Enterobacteriaceae* [1]. In particular, *Klebsiella pneumoniae* carbapenemase (KPC)-2 is the most prevalent carbapenemase in *K. pneumoniae* isolates from South Korea [2]. ST11 is the main clone of KPC-2-producing *K. pneumoniae* isolates from South Korea [3,4]. However, a recent outbreak of ST307 isolates was reported, and its dissemination was identified in South Korea [5,6].

Several virulence factors such as capsule, lipopolysaccharide, siderophores, and fimbriae were identified in *K. pneumoniae* [7], and pathogenicity varies from clone to clone [8].

In our previous study, we identified two main clones of KPC-2-producing *Klebsiella pneumoniae* isolates from a hospital in South Korea, namely, ST11 and ST307 [9], which showed variations in resistance profiles for some antibiotics, such as gentamicin and trimethoprim-sulfamethoxazole. In addition, ST11 appeared to be more virulent than ST307. However, virulence tests were only performed for each of the two carbapenem-resistant isolates, limiting the reliability of the conclusion.

In this study, we investigate the characteristics of the two clones by expanding the number of isolates, including carbapenem-susceptible *K. pneumoniae*. We found that virulence may vary depending on the serotype, even within a particular clone.

## 2. Materials and Methods

### 2.1. K. pneumoniae Isolates and Genotyping

A total of 38 *K. pneumoniae* isolates from South Korea are examined in this study: 18 belonging to ST11 and 20 belonging ST307. Of the ST11 and ST307 isolates, 8 and 10 isolates were carbapenem-resistant, respectively. All the carbapenem-resistant isolates produced KPC-2, which was detected via polymerase chain reaction (PCR) amplification and Sanger sequencing [10]. Other metallo-β-lactamases were not identified (Appendix A). Genotypes of *K. pneumoniae* isolates were determined using the multilocus sequence typing method, as described previously [11]. Capsular polysaccharide types were identified using *wzi* sequencing [12]. The presence of the accessory gene-encoded virulence factors *rmpA*, *kfu*, *iro*, *ybtS*, *iutA*, *allS*, and *clbB* was detected using PCR (Appendix A) [13,14,15].

### 2.2. Antibiotic Susceptibility Testing and String Test

In vitro antibiotic susceptibility testing was performed on all *K. pneumoniae* isolates using the broth microdilution method according to the Clinical and Laboratory Standard Institute (CLSI) guidelines [16]. Ten antimicrobial agents were tested: imipenem, meropenem, cefepime, trimethoprim–sulfamethoxazole, aztreonam, kanamycin, gentamicin, ciprofloxacin, colistin, and tigecycline. CLSI susceptibility breakpoints were employed for all antimicrobial agents except tigecycline. For tigecycline, the Food and Drug Administration breakpoints were used: susceptible, minimal inhibitory concentration (MIC), ≤2 mg/L; intermediate, MIC, 4 mg/L; and resistant, MIC, ≥8 mg/L. *E. coli* ATCC 25922 and *Pseudomonas aeruginosa* ATCC 27853 were used as control strains.

The hypermucoviscosity of *K. pneumoniae* isolates was evaluated using the string test [17]. Overcultured bacterial colonies were plated on blood agar plates using an inoculation loop, and isolates that formed viscous strings >5 mm in length were considered to be hypermucoviscous.

### 2.3. Serum Resistance Assay

For all *K. pneumoniae* isolates included in this study, a serum resistance assay was performed as described previously with slight modifications [18]. Normal human serum (NHS; Innovative Research, Novi, MI, USA) was used to treat mid-log phase bacterial cultures, and heat-inactivated human serum (HIS) was used as a control to determine the bactericidal effect of NHS. After 3 h of incubation at 37 °C with shaking, the mixtures were serially diluted and plated on blood agar. The number of colony-forming units (CFUs) that survived after NHS treatment was compared with that of CFUs that survived after HIS treatment. All assays were performed in triplicate.

### 2.4. Macrophage Infection Assay

The virulence of *K. pneumoniae* in mammalian cells was examined using mouse macrophage cell line RAW 264.7 as described previously [9]. The cells were cultured in Dulbecco’s Modified Eagle’s Medium (Corning, Coring, NY, USA) supplemented with 10% fetal bovine serum (Cytiva, Marlborough, MA, USA), 1% antibiotic–antimycotic solution (Gibco, Waltham, MA, USA), and 1% L-glutamine (Lonza, Basel, Switzerland). The cells were then seeded in a 24-well cell culture plate (SPL Life Sciences, Gyeonggi-do, Korea) at a density of 5 × 105 cells per well and incubated for 20 h prior to bacterial infection. Duplicate RAW 264.7 monolayers were challenged with each *K. pneumoniae* isolate in two replicate 24-well plates. Overnight cultured bacteria were adequately diluted to infect RAW 264.7 cells at a multiplicity of infection of 20. Infected macrophages were incubated for 30 min to permit phagocytosis. The cell culture media were replaced with media that additionally contained 150 mg/L gentamicin to eliminate extracellular bacteria, followed by incubation for 1 h. After incubation, one of the replicate plates was washed with Dulbecco’s Phosphate-Buffered Saline (DPBS; Welgene, Gyeongsan, Korea) and lysed with 1% Triton X-100/DPBS. Meanwhile, the other plate was further incubated with media containing 15 mg/L gentamicin for 20 h, and then washed and lysed with 1% Triton X-100/DPBS. Lysates were sufficiently diluted, and 10 μL of each diluent was dropped onto Luria-Bertani agar plates for measurement of bacterial cells. Macrophage-internalized bacteria were estimated by dividing the CFU of lysates without the additional 20 h incubation with that of lysates with further incubation.

### 2.5. Statistical Analysis

Statistical analysis was performed using GraphPad Prism (version 8.3.0; GraphPad Software, San Diego, CA, USA). The chi-squared test was used to evaluate the difference in the antibiotic susceptibility test. Student’s *t*-test and analysis of variance (ANOVA) with Bonferroni’s multiple-comparison test were used to compare differences in serum resistance and macrophage internalization. Statistical significance was set at *p* < 0.05 (* *p* < 0.05 and ** *p* < 0.01).

## 3. Results

In this study, we included carbapenem-susceptible *K. pneumoniae* isolates of ST11 and ST307 in addition to other carbapenem-resistant isolates. As in the previous study, ST11 isolates had heterogeneous *wzi* alleles (i.e., serotypes; Table 1). *wzi* typing was used for K-serotyping. Although all K types could not be deduced by *wzi* sequences, *wzi* sequencing represents a powerful strain typing method in *K. pneumoniae*. Although all ST307 carbapenem-resistant and -susceptible isolates showed the *wzi173* type, ST11 isolates had three main *wzi* alleles, namely, *wzi14* (K14; five isolates), *wzi50* (six isolates), and *wzi39* (K39; five isolates), and two minor *wzi* alleles, namely, *wzi75* (KL105) and *wzi123* (KL136). The *wzi50*-type alleles were found in both carbapenem-resistant and -susceptible ST11 isolates. In contrast, *wzi14*-type alleles were identified only in carbapenem-resistant isolates, and *wzi39*-type alleles were identified only in carbapenem-susceptible isolates. In one ST307 carbapenem-resistant isolate (SCH2106-08), a nonsense mutation of *wzi* (Gln20*) was found resulting in premature termination. No hypermucoviscosity was observed in all *K. pneumoniae* isolates included in this study, as revealed by the string test.

Antibiotic susceptibility was not different between the two clones (Appendix A). The resistance rates to trimethoprim–sulfamethoxazole (77.8% vs. 95.0%), kanamycin (38.9% vs. 55.0%), and gentamicin (33.3% vs. 50.0%) were higher in ST307 than those in ST11, but were not significantly different. 

Regarding the virulence genes, ST307 isolates showed very simple distribution. In the ST307 isolates, except for the four carbapenem-susceptible ones, only *ybtS* encoding yersiniabactin was detected (Table 1). In contrast, ST11 isolates had a wide distribution of virulence genes. *ybtS* was identified in 12 ST11 isolates; seven out of eight ST11 carbapenem-resistant isolates possessed it. *rmpA2* and *iutA*, which encode a regulator of mucoid phenotype genes and an aerobactin receptor, respectively, were found simultaneously in three ST11 carbapenem-resistant isolates. All three isolates positive to *rmpA2* and *iutA* represented the *wzi14*-type allele. However, a frameshift mutation in *rmpA2*, Asn118Thr, was identified in the three *rmpA2*-positive isolates, leading to premature termination. This may explain the nonmucoid phenotype displayed by the isolates despite the presence of *rmpA2*. *clbB*, which encodes colibactin, was detected in the other four ST11 isolates. Meanwhile, *kfu* and *iro*, which are associated with ferric iron uptake and salmochelin production, respectively, were not detected in any isolates. 

The survival rates against human serum were also highly diverse in ST11 isolates, whereas ST307 had a relatively narrow range of survival rates (Figure 1A). The survival rates against human serum were not significantly different between the two clones. However, different results were obtained when the ST11 isolates were divided according to *wzi* allele type. The ST11 isolates with *wzi14*-type allele showed a significantly higher survival rate against human serum than those with *wzi50*-type and *wzi39*-type alleles (Figure 1B).

Internalization rates into the macrophage using mouse macrophage cell line RAW 264.7 were measured. As in the survival rates against human serum, internalization rates were more heterogeneous in ST11 isolates than those in ST307 isolates (Figure 2A). The survival rates of ST307 were lower than those of ST11 isolates, possibly indicating higher virulence of ST307 isolates, although the difference was not significant. ST11 isolates with the *wzi14*-type allele showed lower internalization rates compared with those with other *wzi* alleles (Figure 2B). 

Furthermore, we analyzed the survival and internalization rates between *ybtS*-positive and *ybtS*-negative isolates, since *ybtS* is distributed in both clones (Figure 3). *ybtS*-positive isolates showed significantly higher survival rates against human serum than those of *ybtS*-negative isolates (*p* value = 0.0188; Figure 3A). In the macrophage infection experiment, fewer internalized cells were *ybtS*-positive isolates than *ybtS*-negative isolates, although the difference was not significant (*p* value = 0.0681; Figure 3B). 

## 4. Discussion

In our previous study, we compared two main clones of KPC-2-producing *K. pneumoniae* isolates form South Korea, ST11 and ST307 [9]. The two clones showed different characteristics. ST11 carbapenem-resistant isolates showed higher MICs for carbapenems. The two clones showed different serotypes and plasmid types. ST11 isolates represent diverse incompatibility types compared to the homogenous plasmids of ST307 KPC-2-producing isolates, supporting that ST307 has been recently introduced in South Korea [6]. We hypothesized that ST11 may be more virulent than ST307 on the basis of the serum-resistance and macrophage-infection assays of the two isolates.

In this study, we investigated more isolates, including carbapenem-susceptible isolates. No difference in virulence related to serum resistance and macrophage internalization was observed between the two clones. In our previous study, the difference between KPC-2-producing ST11 and ST307 isolates might have been due to selection bias [9]. However, survival rates against human serum and internalization rates of ST11 isolates were diverse, which was shown in other features including the serotype.

Further analysis of ST11 isolates by serotype revealed differences in virulence. Specifically, *wzi14*-type or serotype K14 showed higher virulence than the other serotypes did. All five *wzi14*-type ST11 isolates contained *ybtS*, three of which possessed additional virulence genes *rmpA2* and *iutA*. Although the high virulence with respect to serum resistance and macrophage internalization would be responsible for the feature of the capsule (i.e., serotype) in *K. pneumoniae* [7], the presence of more virulence genes also contributes to the pathogenicity of the isolates. In our analysis, the *ybtS*-positive isolates had higher serum resistance-related virulence than that of the *ybtS*-negative isolates.

Overall, multiple serotypes of the ST11 clone means that various capsular polysaccharide loci were transferred into the isolates, suggesting that this clone has existed and disseminated not just recently, at least in South Korea. In contrast, ST307 was identified as a clone that had recently been introduced into South Korea, regardless of whether KPC-2 is produced or not. Notably, the virulence of the ST307 isolates was as high as that of the *wzi14*-type isolates of ST11. Although the KPC-2-producing ST307 isolate may have been introduced independently from the susceptible ST307 isolate, the possibility that the plasmid bearing blaKPC-2 was introduced into the susceptible ST307 isolate after it had been introduced into South Korea cannot be excluded. Recently, ST307 carbapenem-resistant *K. pneumoniae* isolates have been globally identified [19,20,21,22], which may have been facilitated by conjugable-plasmid-bearing *bla*_KPC-2_ [23]. Global genomic studies of ST307 isolates, including carbapenem-susceptible and -resistant isolates, would reveal the evolution of this recently emerged high-risk *K. pneumoniae* clone.

In this study, we compared the virulence of two main clones of *K. pneumoniae* isolates from South Korea. ST11 isolates had diverse features including serotypes, and a particular serotype, *wzi14*, showed higher virulence than other serotypes of the clone did. Meanwhile, ST307 isolates exhibited homogeneous features and a similar virulence level as that of the *wzi14*-type ST11 isolates. Continuous monitoring of the transmission of these two major clones is needed, and research on the elaboration of treatments including antibiotic selection according to the characteristics of each clone is required.

## Figures and Tables

**Figure 1 microorganisms-10-01827-f001:**
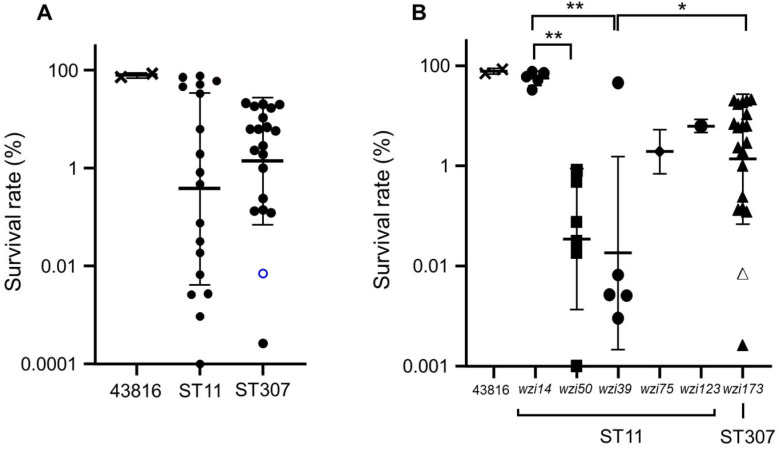
Results of serum resistance assay. (**A**) Survival rates of ST11 and ST307 isolates were compared using Student’s *t*-test. (**B**) Survival rates were compared among the *wzi* allele types using analysis of variance (ANOVA) with Bonferroni’s multiple-comparison test. The (**A**) blank circle and (**B**) blank triangle in ST307 represent the survival rate of SCH2106-08, in which a C58T mutation leading to Q20* in the *wzi* gene was detected. * *p* < 0.05, ** *p* < 0.091.

**Figure 2 microorganisms-10-01827-f002:**
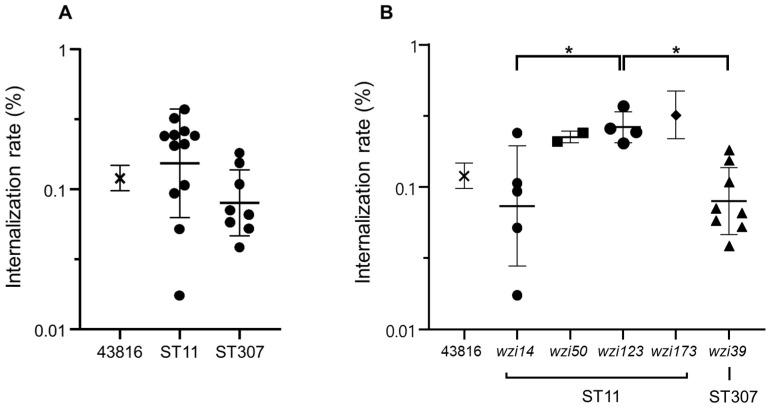
Results of the macrophage infection experiment. Only gentamicin-susceptible isolates were included in this experiment, as gentamicin was used to eliminate extracellular bacteria. (**A**) Macrophage internalization rates in ST11 and ST307 isolates analyzed using Student’s *t*-test. (**B**) Internalization rates were compared among the *wzi* allele types using ANOVA with Bonferroni’s multiple-comparison test. * *p* < 0.05.

**Figure 3 microorganisms-10-01827-f003:**
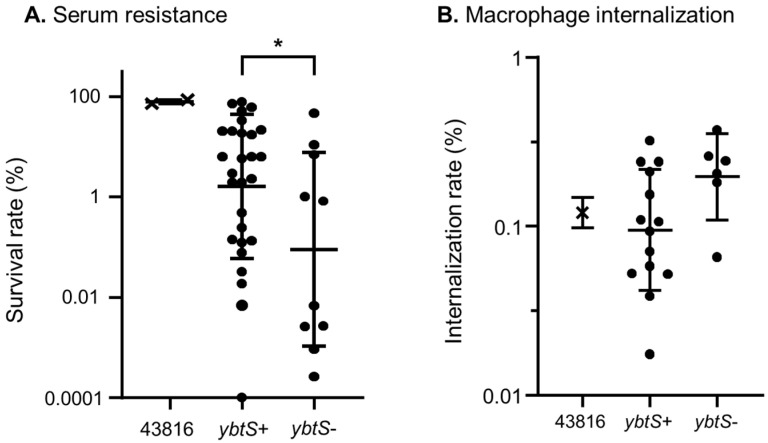
(**A**) Serum resistance and (**B**) macrophage internalization were compared between the *ybtS*-positive and -negative isolates regardless of clones. Student’s *t*-test was used for statistical analysis. * *p* < 0.05.

**Table 1 microorganisms-10-01827-t001:** Sources, *wzi* alleles, and virulence genes of two *K. pneumoniae* clones from South Korea, ST11 and ST307.

Genotype	Carbapenem	Isolate Number	Source	MBL and ESBL	*wzi* Allele	Virulence Genes
ST11	Carbapenem-resistant	SCH2104-31	Bile	KPC-2	14	*rmpA2*, *ybtS*, *iutA*
		SCH2104-32	Voided urine	KPC-2, CTX-M-15	14	*rmpA2*, *ybtS*, *iutA*
		SCH2106-16	Rectal swab	KPC-2	50	*ybtS*
		SCH2107-22	Catheterized urine	KPC-2, CTX-M-15	50	*ybtS*
		SCH2108-30	Sputum	KPC-2	14	*rmpA2*, *ybtS*, *iutA*
		SCH2104-17	Catheterized urine	KPC-2	14	*ybtS*
		SCH2108-43	Sputum	KPC-2	50	
		SCH2108-44	Bronchial washing	KPC-2	75	*ybtS*
	Carbapenem-susceptible	SCH2107-06	Bile	CTX-M-9	50	*ybtS*, *clbB*
		SCH2108-31	ND ^a^	CTX-M-15	14	*ybtS*
		B0706-169	ND	CTX-M-15	50	*ybtS*, *clbB*
		B0708-216	ND	CTX-M-15	39	
		K01-Bact-08-03094	ND	CTX-M-15	39	
		K01-Bact-08-10058	ND	CTX-M-15	50	*ybtS*, *clbB*
		K01-Bact-08-12164	ND	CTX-M-15	39	
		K01-Bact-08-12216	ND	CTX-M-15	39	
		K01-Bact-08-12226	ND	CTX-M-15	39	
		726	Urine	CTX-M-15	123	*ybtS*, *clbB*
ST307	Carbapenem-resistant	SCH2101-15	Ascitic fluid	KPC-2, CTX-M-15	173	*ybtS*
		SCH2102-16	Bile	KPC-2, CTX-M-15	173	*ybtS*
		SCH2102-30	Rectal swab	KPC-2, CTX-M-15	173	*ybtS*
		SCH2104-07	Voided urine	KPC-2, CTX-M-15	173	*ybtS*
		SCH2104-33	Rectal swab	KPC-2, CTX-M-15	173	*ybtS*
		SCH2105-10	Catheterized urine	KPC-2, CTX-M-15	173	*ybtS*
		SCH2105-20	Rectal swab	KPC-2, CTX-M-15	173	*ybtS*
		SCH2106-08	Rectal swab	KPC-2, CTX-M-15	173 ^b^	*ybtS*
		SCH2108-07	Blood	KPC-2, CTX-M-15	173	*ybtS*
		SCH2109-15	Sputum	KPC-2, CTX-M-15	173	*ybtS*
	Carbapenem-susceptible	SCH2012-07	Blood	CTX-M-15	173	*ybtS*
		SCH2012-19	Blood	CTX-M-15	173	*ybtS*
		SCH2107-01	Blood	CTX-M-15	173	*ybtS*
		SCH2107-07	Blood	CTX-M-15	173	*ybtS*
		SCH2107-08	Blood	CTX-M-15	173	
		SCH2107-19	Blood	None	173	
		925	Peritoneal fluid	None	173	*ybtS*
		SCH-CR31	Peritoneal fluid	None	173	*ybtS*
		633	Stool	None	173	
		SCH2106-18	Blood	None	173	

^a^ ND, not described. ^b^ C58T mutation led to Q20*, resulting in premature stop in *wzi* gene.

## Data Availability

Not applicable.

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
