# Peer review of "Comparison of Virulence between Two Main Clones (ST11 and ST307) of Klebsiella pneumoniae Isolates from South Korea"

_microorganisms, 2022, doi:10.3390/microorganisms10091827_

Round 1
Reviewer 1 Report
Dear authors
The manuscript is well structured, needs adjustment.
I believe that editing and rewriting same part of the text is necessary to make the manuscript easily readable, but this effort would be well worth it as the authors have included a great deal of information.
However, I have following comments that should be addressed:
- Please recheck the References order
- Double Check abreviation!
- Table 1 is hard to read can be reformat!
In the introduction part first part need to be rewrite.
The article need to be double check by an native English speaker.
Please streamline 3 short conclusion! The conclusion need to focus on clinical improvement of patients treatment, and to introduce in clinical use.
The conclusion part need more explanation and clear points.
Author Response
Please recheck the References order
- We rechecked and confirmed it.
Double Check abreviation!
- As suggested, we recheck it.
Table 1 is hard to read can be reformat!
- We tried to reformat the Table 1 in several ways, but nothing seemed better than the current one. We think the current format is best.
In the introduction part first part need to be rewrite.
- As suggested, we rewrote the first paragraph in the Introduction.
“Carbapenem resistance by carbapenemase has caused antibiotic treatment failure and has threatened public health worldwide in Enterobacteriaceae [1]. In particular, Klebsiella pneumoniae carbapenemase (KPC)-2 is the most prevalent carbapenemase in K. pneumoniae isolates from South Korea [2]. ST11 has been the main clone of KPC-2-producing K. pneumoniae isolates from South Korea [3, 4]. However, an recent outbreak of the ST307 isolates has been reported and its dissemination has been identified in South Korea [5, 6]. (Line 27-33 in the revised manuscript)
The article need to be double check by an native English speaker.
- A native English speaker corrected the paper. Attached is a certificate of this.
Please streamline 3 short conclusion! The conclusion need to focus on clinical improvement of patients treatment, and to introduce in clinical use.
The conclusion part need more explanation and clear points.
- As suggested, we revised and supplemented the conclusions.
“In this study, we compared the virulence of two main clones of K. pneumoniae isolates from South Korea. ST11 isolates had diverse features including serotypes, and a particular serotype, wzi14, showed higher virulence than other serotypes of the clone. Meanwhile, ST307 isolates exhibited homogeneous features and a similar virulence level as that of the wzi14-type ST11 isolates. Continuous monitoring of the transmission of these two major clones is needed, and research on the elaboration of treatments including antibiotic selection according to the characteristics of each clone is required.” (Line 220-226 in the revised manuscript)
Reviewer 2 Report
This is an interesting study in which the virulence of two main clones of K. pneumoniae isolates from South Korea, ST11 and ST307 was compared, including carbapenem-resistant and -susceptible isolates. Considering the pathogenicity of this bacterium and its global distribution, it is important to continuously characterize the isolated variants.
Some suggestions for improvement:
Line 26: The “Introduction” should be improved by adding more information about the pathogenicity and the genetic variability of K. pneumoniae
Line 29: use italics Enterobacteriaceae
Line 51: use italics for K. pneumoniae
Line 108 & 119: same
Line 110: It would be useful for the reader to provide some information on the significance of wzy allele typing for Klebsiella pneumoniae
Line 110: wzy in italics
Lines 124-135: all names of genes should be in italics
Line 127: please provide also the name of the product encoded by the gene ybtS
Line 135: Also for kfu and iro
Author Response
Reviewer 2
Line 26: The “Introduction” should be improved by adding more information about the pathogenicity and the genetic variability of K. pneumoniae
- As suggested, we added the information on the pathogenicity and genetic variability in K. pneumoniae.
“Several virulence factors such as the capsule, lipopolysaccharide, siderophores, and fimbriae have been identified in K. pneumoniae [7], and it is known that pathogenicity varies from clone to clone [8].” (Line 34-36 in the revised manuscript)
Line 29: use italics Enterobacteriaceae
- As suggested, we italicized it.
Line 51: use italics for K. pneumoniae
- We italicized it.
Line 108 & 119: same
- I don't know what the reviewer mean.
Line 108, “and ST307, in addition to other carbapenem-resistant isolates. As in the previous study,
Line 119, “test.”
- If the reviewer’s comment in on the italicitation of K. pneumoniae, we italicized them.
Line 110: It would be useful for the reader to provide some information on the significance of wzi allele typing for Klebsiella pneumoniae
- As suggested, we mentioned it.
“wzi typing has been used for K-serotyping. Although all K types could not deduced by wzi sequences, wzi sequencing represents a powerful strain typing method in K. pneumoniae.” (Line 112-114 in the revised manuscript)
Line 110: wzi in italics
- We italicized it.
Lines 124-135: all names of genes should be in italics
- We italicized them.
Line 127: please provide also the name of the product encoded by the gene ybtS
- It has already been mentioned that ybtS encodes yersiniabactin in the original manuscript.
Line 135: Also for kfu and iro
- We mentioned the product of them.
“Meanwhile, kfu and iro, which are associated with ferric iron uptake and salmochelin production, respectively, were not detected in any isolates.” (Line 140-141 in the revised manuscript)